# Perioperative and Oncological Outcomes of Robotic-Assisted, Video-Assisted Thoracoscopic and Open Lobectomy for Patients with N1-Metastatic Non-Small Cell Lung Cancer: A Propensity Score-Matched Study

**DOI:** 10.3390/cancers14215249

**Published:** 2022-10-26

**Authors:** Hanbo Pan, Yu Tian, Hui Wang, Long Jiang, Zenan Gu, Hongda Zhu, Junwei Ning, Jia Huang, Qingquan Luo

**Affiliations:** Department of Thoracic Surgical Oncology, Shanghai Chest Hospital, Shanghai Jiao Tong University School of Medicine, Shanghai 200030, China

**Keywords:** non-small cell lung cancer, perioperative outcomes, oncological outcomes, robotic-assisted thoracoscopic lobectomy, metastatic N1 lymph nodes, propensity score-matched analysis

## Abstract

**Simple Summary:**

Non-small cell lung cancer (NSCLC) is one of the most prevalent and deadly malignancies worldwide, and patients with metastatic N1 lymph nodes (LNs) are associated with a worse prognosis. Despite the fact that robotic-assisted thoracoscopic lobectomy (RATL) has been prevalently applied in treating early stage NSCLC, its advantages for patients with involved N1 LNs remain unknown. This retrospective study compared perioperative and oncological outcomes among RATL, video-assisted thoracoscopic lobectomy (VATL), and open lobectomy (OL) in a cohort of 855 consecutive cases with pathological N1 NSCLC, thereby aiming to assess the superiority of RATL over traditional surgical approaches for NSCLC patients with metastatic N1 LNs. RATL resulted in the most optimal surgical outcomes, the fastest recovery, and the lowest morbidities of postsurgical complications among the three surgical methods, and also assessed more N1 and total LNs and led to a higher incidence of postoperative upstaging than VATL, though it achieved comparable oncological outcomes in relation to VATL and OL.

**Abstract:**

(1) Background: Despite the fact that robotic-assisted thoracoscopic lobectomy (RATL) has been prevalently applied for early stage non-small cell lung cancer (NSCLC), its superiorities are still to be fully revealed for patients with metastatic N1 lymph nodes (LNs). We aim to evaluate the advantages of RATL for N1 NSCLC. (2) Methods: This retrospective study identified consecutive pathological N1 NSCLC patients undergoing RATL, video-assisted thoracoscopic lobectomy (VATL), or open lobectomy (OL) in Shanghai Chest Hospital between 2014 and 2020. Further, perioperative and oncological outcomes were investigated. (3) Results: A total of 855 cases (70 RATL, 435 VATL, and 350 OL) were included. Propensity score matching resulted in 70, 140, and 140 cases in the RATL, VATL, and OL groups, respectively. RATL led to (1) the shortest surgical time (*p* = 0.005) and lowest intraoperative blood loss (*p* < 0.001); (2) the shortest ICU (*p* < 0.001) and postsurgical hospital (*p* < 0.001) stays as well as chest tube duration (*p* < 0.001); and (3) the lowest morbidities of postsurgical complications (*p* = 0.016). Moreover, RATL dissected more N1 (*p* = 0.027), more N1 + N2 (*p* = 0.027) LNs, and led to a higher upstaging incidence rate (*p* < 0.050) than VATL. Finally, RATL achieved a comparable 5-year disease-free and overall survival in relation to VATL and OL. (4) Conclusions: RATL led to the most optimal perioperative outcomes among the three surgical approaches and showed superiority in assessing N1 and total LNs over VATL, though it did achieve comparable oncological outcomes in relation to VATL and OL for N1 NSCLC patients.

## 1. Introduction

Lung cancer (LC) is one of the most prevalent malignancies and the top cause of tumor-related deaths globally; further, non-small cell lung cancer (NSCLC) accounts for approximately 85% of general LC morbidities [1]. Although great improvements have been made in the diagnosis of early stage NSCLC, most cases are initially diagnosed with the locally advanced disease, and patients with N1 lymph node (LNs) metastasis are associated with a remarkably worse prognosis with their 5-year overall survival (OS) at about 50% than that without nodal involvement, which results in a 5-year OS of approximately 85% [2,3,4,5]. Despite open lobectomy (OL) still being the standard surgical procedure, video-assisted thoracoscopic lobectomy (VATL), a minimally invasive surgery (MIS) technique, has also been widely used for locally advanced NSCLC [6]. Multiple studies have indicated that VATL could reduce perioperative complications and short-term morbidities and also achieve comparable long-term outcomes to OL for NSCLC patients with positive N1 and/or N2 LNs, suggesting that MIS may be a favorable surgical approach for locally advanced NSCLC [7,8].

Nowadays, robotic-assisted thoracoscopic lobectomy (RATL), an innovative MIS technique that has attracted the growing interest of thoracic surgeons, has been accepted to be feasible and oncologically effective in treating early stage NSCLC [9,10]. RATL provides a magnified, 3-dimensional (3D) visualization that allows surgeons to perform complicated surgery precisely and has shown the advantages of shorter surgical durations, decreased postoperative pains, and enables faster recoveries when compared with VATL and OL [11]. More importantly, the robot-assisted operation system possesses a highly flexible mechanical wrist of which its maneuverability is even superior to human hands, thereby making radical lymphadenectomy more convenient [12]. Therefore, RATL might be especially suitable for NSCLC patients with lymph nodal metastasis. Previous studies have indicated RATL to be safe and effective for IIIA-N2 NSCLC and may even provide short- and long-term benefits when compared with OL [13,14,15]. However, the comparison of RATL versus traditional surgical approaches concerning the perioperative and long-term outcomes specified in treating N1 NSCLC patients is limited, and the superiority of RATL for this important group of patients remains unrevealed.

In the present study, we retrospectively evaluated the perioperative and oncological outcomes of RATL, VATL, and OL for NSCLC patients with pathological-confirmed N1 lymph nodal metastasis, aiming to investigate the superiority of RATL in treating N1 NSCLC patients. Propensity score matching (PSM) was adopted in order to reduce the potential bias in patient selection.

## 2. Materials and Methods

### 2.1. Study Design

The present study is a single-center, retrospective comparative cohort study of NSCLC patients with N1 lymph nodal metastasis who underwent lobectomy at Shanghai Chest Hospital, Shanghai Jiao Tong University School of Medicine, and was permitted by the IRB of Shanghai Chest Hospital (No. KS1735). All procedures conducted in the study that related to human participants were performed as per the criteria outlined in the Declaration of Helsinki.

### 2.2. Cases Selection and Data Collection

Consecutive NSCLC patients with metastatic N1 LNs confirmed by pathology reports undergoing lobectomy, between October 2014 and June 2020, were retrospectively identified (Figure 1). Patients receiving RATL before October 2014 were not included in order to help minimize the potential bias due to the learning curve of RATL. Meanwhile, patients undergoing surgery after June 2020 were also excluded because of the relative short-term follow-up. Preoperative examinations such as pulmonary functional testing, echocardiography, and electrocardiogram were performed in order to assure the operation tolerability of patients. Distant metastasis was assessed by applying cranial enhanced magnetic resonance imaging, bone scintigraphy, and positron emission tomography/computed tomography (PET/CT). Mediastinal and pulmonary LN statues were conventionally evaluated by enhanced thoracic CT, while PET/CT, mediastinoscopy, and/or endobronchial ultrasound-guided trans-bronchial needle aspiration were further used if a chest CT scan indicated that the short-axis was larger than 1 cm of a LN. The inclusion criteria were as follows: (1) Instances of RATL, VATL, or OL that were conducted and combined with systemic LN dissection; (2) presence of a pathology report for NSCLC and positive N1 LNs after surgery. The exclusion criteria were as follows: (1) The surgery conducted was not for lung cancer and were not cases with incomplete information; (2) the patient underwent partial resection, segmentectomy, or sleeve resection; (3) underwent pneumonectomy, bi-lobectomy, or bilateral operations; (4) were with a histology other than NSCLC; (5) possessed metastatic lung tumors; (6) were without a systemic ipsilateral pulmonary and a mediastinal lymph node dissection; (7) possessed intrapulmonary or distant metastasis; (8) or underwent neoadjuvant therapy.

It must be noted, that the surgical approach used was decided by both patients and surgeons together prior to the surgery. A total of 855 consecutive N1 NSCLC patients were enrolled and further classified into the RATL, VATL, and OL groups according to the surgical approach each patient received. The following data of the included patients were recorded, including: (1) Clinicopathological features such as age, sex, body mass index (BMI), smoking status, preoperative comorbidities, pulmonary functions, location, histological type, size, visceral pleural invasion, pathological stage of the primary tumor, pathological TNM stage of the disease, and adjuvant therapy; (2) surgical outcomes such as resection margin, surgical duration, intraoperative blood loss, and blood transfusion; (3) postoperative recoveries including length of ICU and postsurgical stays, duration and volume of chest tube drainage; (4) postsurgical complications; (5) LN assessments, such as: N1 LN assessments including N1 LN and stations counts, hilar (#10), interlobar (#11), and lobar (#12) LN counts, and LN assessing frequency, N2 LN assessments including N2 LN and stations counts, as well as lymph nodal upstaging; (4) oncological outcomes such as the 1-year, 3-year, and 5-year disease-free survival (DFS) and OS. All cases were staged according to the tumor, node, and metastasis (TNM) staging system (8th edition) of IASLC.

### 2.3. Surgical Procedures

RATL, VATL, and OL were all performed following the procedures that was reported by our team in previous studies [13,15,16]. Briefly, all patients received a radical lobectomy combined with systemic LN assessment; further, the mediastinal and hilar LNs were regularly harvested. RATL was conducted by adopting the da Vinci Surgical System (Intuitive Surgical, CA, USA) according to the definition of the Cancer and Leukemia Group B 39802. RATL and VATL were performed through four minimal incisions with a non-rib-spreading technique. Further, patients in the OL group underwent a routine rib-spreading thoracotomy via an incision of 15–20 cm. The intraoperative rapid frozen section was required for every patient during the operation.

### 2.4. Postoperative Management and Follow-Up

The enhanced postoperative recovery protocol including preoperative smoking withdrawal and breathing training, as well as early postoperative activities and extraction of the chest tube, were routinely applied to all patients. The standard adjuvant therapy such as chemotherapy, radiotherapy, chemoradiotherapy, target therapy, or immunotherapy was recommended for every patient depending on the individual circumstance, unless the therapy was unfeasible. 

After discharge from the hospital, patients were reviewed via using a brain MRI and thoracic CT scan every 3 months during the first two years and every year afterward. The telephone and/or internet follow-up was conducted once a year until death, or January 2022 for the patients who did not periodically come to outpatient care. Cases that missed follow-up were assessed according to the most recent medical records.

### 2.5. Statistical Analysis

Statistical analysis was conducted following the previously described methods [17,18]. Appropriate descriptive statistics were adopted to express variables, including median [range] or mean ± standard deviation (SD) for the continuous variable and frequencies (percentages) for the categorical variable. For the continuous variable, the Kolmogorov–Smirnov test was adopted in order to analyze the homogeneity of variance and the normality of distribution. For variables with normal distribution and homogeneous variance, the analysis of variance followed by Tukey’s multiple comparisons test was performed. Otherwise, the Kruskal–Wallis rank sum test, followed by Dunn’s multiple comparisons test, was conducted. Pearson’s χ^2^ or Fisher exact test followed by the Bonferroni post hoc test was used to compare the categorical variable. The Kaplan–Meier curves log-rank (Mantel–Cox) test was adopted to analyze the oncological outcomes, and the multiple Cox regression model analysis was further applied to analyze factors relevant to DFS and OS. Statistical analysis was performed using SPSS 26.0 (IBM Corporation, Armonk, NY, USA). Oncological outcomes were analyzed using GraphPad Prism 9 (GraphPad Software Inc., San Diego, CA, USA). The *p*-value < 0.050 was considered statistically significant.

To minimize the potential bias in case selection, propensity score matching (PSM), applying the nearest matching method, was used to achieve equilibrium with the baseline confounding features of included cases with a 1:2:2 RATL versus VATL versus OL group ratio. The included cases were matched with the 8 variables, which were as follows: (1) categorical variables including sex, histological type, and pathological TNM stage; (2) continuous variables including age, BMI, DLCO %, FEV1 %, and tumor size. PSM was performed by applying the R version 4.1.3 (The R Foundation for Statistical Computing, Vienna, Austria).

## 3. Results

### 3.1. Clinicopathologic Characteristics

The baseline demographic and clinicopathologic features of patients before PSM are summarized in Table 1. The OL group possessed the highest percentage of males (*p* < 0.001) and the largest tumor size (*p* < 0.001) among the three groups. The RATL, VATL, and OL groups also remarkably differed in smoke status (*p* < 0.001), histological type (*p* < 0.001), pathological T (*p* < 0.001), and TNM (*p* < 0.001) stages. PSM was then applied to achieve equilibrium with the baseline clinicopathologic features among the RATL, VATL, and OL groups; as such, 350 patients were finally identified. As expressed in Table 2, patients in the three groups had comparable distributions of all included baseline clinicopathologic characteristics following PSM.

### 3.2. Perioperative Outcomes

The perioperative outcomes of the RATL, VATL, and OL groups after PSM are expressed in Table 3. In terms of surgical outcomes, RATL led to the shortest surgical time (*p* = 0.005) and the lowest intraoperative blood loss (*p* < 0.001) among the three groups. The three surgical approaches led to a comparable resection margin (*p* = 0.330), reoperation incidence (*p* = 0.780), and intraoperative blood transfusion rate (*p* = 0.844). Further multiple comparisons showed that RATL was associated with a significantly shorter operation duration than VATL (*p* = 0.003) and notably fewer intraoperative blood loss than OL (*p* < 0.050). Moreover, the RATL group had the shortest length of ICU (*p* < 0.001) and postsurgical (*p* <0.001) stays as well as chest tube drainage duration (*p* < 0.001). The three groups had comparable chest tube drainage volumes (*p* = 0.967). Further multiple comparisons showed that RATL led to a significantly shorter length of ICU (*p* < 0.001) and postsurgical (*p* < 0.001) stays as well as a notably shorter chest tube duration (*p* < 0.001) than OL. Finally, RATL led to the fewest postsurgical comorbidities, followed by VATL, and OL (*p* = 0.016). Further multiple comparisons revealed that RATL led to significantly fewer postsurgical comorbidities than OL (*p* < 0.050).

### 3.3. LNs Assessment

The LN assessment of RATL, VATS, and OL after PSM is shown in Table 4. OL dissected the most N1 (*p* = 0.013) and also total (*p* = 0.001) LNs. Further multiple comparisons showed that RATL harvested significantly more N1 (*p* = 0.027) and N1 + N2 LNs (*p* = 0.027) when compared with VATL and assessed comparable N1 (*p* = 1.000) and total (*p* = 0.950) LNs to OL. Moreover, RATL was associated with the highest incidence of postoperative upstaging (*p* = 0.040), and further multiple comparisons showed that the incidence of postoperative upstaging of the RATL group was also notably higher than that of the VATL group (*p* < 0.050) and comparable to that of the OL group (*p* > 0.050). Finally, the three groups had a similar (1) frequency and overall and positive count of assessed #10, #11, and #12 LNs; (2) overall and positive count of dissected N1 stations and positive count of assessed N1 LNs; (3) overall count of harvested N2 LNs and stations; and (4) overall count of dissected N1 + N2 stations (all *p* > 0.050).

### 3.4. Oncological Outcomes

After PSM, the overall median postoperative follow-up was 41.5 months [interquartile range (IQR), 26.0–57.0 months]. For the RATL, VATL, and OL groups, the median follow-up was 39.5 months [IQR, 19.5–59.0 months], VATL 44.5 months [IQR, 30.0–56.0 months], and OL 42.0 months [IQR, 25.0–58.5 months], respectively. In the RATL, VATL, and OL groups, the 5-year DFS was 54.41%, 49.31%, and 51.44%, respectively, and the 5-year OS was 68.59%, 62.81%, and 66.24%, respectively (Figure 2A,B). The three surgical approaches were associated with the similar DFS (*p* = 0.599) and OS (*p* = 0.765) data. In addition, the subgroup analysis indicated no survival data difference among the RATL, VATL, and OL groups in terms of the tumor size (Figure 3A–D) and the pathological TNM stage (Appendix A, Figure A1A–D). Furthermore, multivariate Cox regression model analysis revealed that the operation approach was not individually associated with the DFS (Hazard Ratio (HR) = 1.008, *p* = 0.970; Table 5) or OS (HR = 0.855, *p* = 0.577), and receiving adjuvant therapy was independently associated with prolonged DFS (HR = 0.520, *p* = 0.012) and OS (HR = 0.405, *p* = 0.003). However, an increased metastatic number of LNs was the independent risk factor of poor DFS (HR = 0.268, *p* < 0.001) and OS (HR = 0.368, *p* < 0.001).

## 4. Discussion

Although RATL has been widely accepted to be safe and effective for lobectomy and is being widely applied in treating early stage NSCLC, its application in advanced-stage NSCLC is still controversial [19]. Several retrospective studies have indicated RATL to be feasible and oncologically effective for the treatment of N2 NSCLC [8,14,20]. Recently, our team conducted a multicenter, randomized, controlled trial, which suggested that RATL improved perioperative outcomes and achieves similar long-term outcomes compared with open thoracotomy for clinical-N2 NSCLC patients [13,15]. However, the study comparing RATL with traditional surgical approaches in perioperative and the oncological outcomes specified for N1 NSCLC patients is limited; further, the superiority of RATL for this important group of patients remains unknown. Although a few retrospective studies have reported perioperative outcomes and LN dissection of RATL specified for N1 NSCLC patients, none of them included the comparison versus VATL and/or OL [3,5,20]. Recently, a retrospective study suggested that RATL led to comparable perioperative outcomes, LN assessment, and short-term survival when compared to VATL for clinical-N1/N2 NSCLC patients [16]. However, this study merely included 57 N1 NSCLC cases. In our retrospective study that identified a total of 855 NSCLC patients with N1 LNs metastasis who underwent RATL, VATL, or OL—RATL was found to result in the best perioperative outcomes among the three surgical approaches and exhibited superiority in N1 LN assessment over VATL, though it achieved comparable oncological outcomes in regard to VATL and OL.

In terms of perioperative outcomes, our results indicated that RATL possessed a significantly shorter surgical time than VATL. Referring to the previous research, RATL was commonly understood to prolong the surgical duration, which may be attributed to the extra docking duration and influence of the learning curve [21,22]. Nevertheless, RATL was usually associated with a shorter surgical duration compared with VATL, according to our previous studies, and the surgical time of RATL in this study was also fewer than that in the research reported by other surgical teams, which may be attributed to the accumulation of surgical experience, a well-organized medical team, and a high-patient-volume medical center [9,16,21,22]. Moreover, RATL led to the fastest postsurgical recoveries and the least postsurgical morbidities in this study, which may be due to the 3D, high-definition visualized surgical field and tremor filtration provided by the robot-assisted operation system that allows operators to perform the surgical resection more precisely and to, therefore, avoid unnecessary damage [23,24].

Despite preoperative LN evaluations having made great improvements, hidden positive N1 LNs are still notable, therefore the sampling of N1 LNs is of critical importance for the surgical treatment of NSCLC patients, especially for those with the clinical-N0 (cN0) disease [25]. Previous studies have evaluated the assessment of N1 LNs by using the robot-assisted surgical system, indicating conflicting results. Three independent studies indicated that RATL harvested more N1 LNs and similar LN stations compared with VATL and OL [26,27,28]. However, Kneuertz et al., demonstrated that RATL failed to show superiority in N1 lymph nodal dissection over VATL or OL [29]. Our results indicated that despite the three surgical approaches dissecting similar LN stations, RATL harvested significantly more total and N1 LNs than VATL, and that such an advantage might be due to the 3D, magnified visualization, and enhanced maneuverability and dexterity of the robot-assisted operation system—which could provide surgeons with improved abilities in dissecting LNs around bronchi and vessels [30]. Postoperative LN upstaging could also present itself as a key indicator of surgical quality due to the limited values of preoperative staging in assessing N1 LNs involvement as in the aforementioned other procedures. According to previous studies, the upstaging incidence varies, ranging from less than 40% to as high as more than 60%, in pathological-N1 (pN1) NSCLC patients [22,31,32,33]. Our results showed that RATL led to the lymph nodal upstaging incidence of 68.57%, which was notably higher than those receiving VATL. Such advantages might be largely attributed to the increased N1 LN assessment by RATL that reduced the incidence of occult N1 LNs metastasis for clinical-N0 NSCLC patients [34]. Although this study merely included pN1 NSCLC patients, our results did reveal that RATL possessed superiority in detecting hidden positive N1 LNs. Therefore, for cN0 NSCLC patients with a high hazard of lymph nodal metastasis—such as the increased level of carcinoembryonic antigen, the high standardized uptake value of PET/CT, and the enlarged tumor size—RATL may be a favorable surgical approach over VATL in the context of leading to more complete staging. This could provide evidence for adjuvant therapy and thus achieve better long-term outcomes [35,36,37]. Moreover, for clinical-N1 NSCLC patients, the breakdown by the pathological N category ranges from approximately 15% to 53%, and the increased LNs sample through RATL may contribute to a more accurate LN staging [20,38,39,40]. Further investigation is needed to verify whether RATL possesses superiority in precise LN staging, and even oncological outcomes over VATL, for clinical-N0 and N1 NSCLC patients with a high hazard of LN involvement. 

In this study, we retrospectively identified pathological N1 NSCLC patients who underwent RATL, VATL, or OL. Further, the different selection tendencies in the surgical approach concerning the clinical N stage of patients may cause selection bias that influenced the conclusion when comparing the lymph nodal upstaging among the three surgical techniques. In our hospital, the option between the MISs and OL approaches mainly depended on the tumor location, while the clinical lymph nodal involvement was relatively less considered. Open surgery was preferred for patients with a central malignancy that had a close relationship with the hilar structures. Therefore, patients who underwent OL were more likely associated with a larger tumor size and advanced T stage, which was evidenced by the patients in the OL group possessing the largest tumor size and the highest proportion of stage IIIA disease before PSM. Meanwhile, NSCLC patients with a history of intrathoracic diseases (e.g., tuberculosis and pleurisy) and/or thoracic surgeries who were estimated to possess a severe pleural adhesion may also undergo OL. In addition, although previous studies have shown the superiority of RATL over VATL in decreasing postoperative pain and dissecting more LNs for resectable NSCLC patients, the selection between RATL and VATL methods was mostly based on the willingness and economic conditions of patients in recent years since RATL is usually associated with increased overall costs and VATL is the most prevalently applied MIS approach, while the clinical lymph nodal metastasis was also less considered [9,22]. Most importantly, the clinical N stage of the enrolled cases was comparable among the RATL, VATL, and OL groups before and after PSM. Taken together, we believed that the bias in the selection of clinical N0 patients among the three surgical approaches was relatively low, and the slight increase in the proportion of clinical N0 patients in the RATL and OL groups over the VATL group was largely attributed to the increased assessment of LNs, which contributed to the discovery of the hidden N1 LNs. 

Due to the relatively short duration that RATL has been applied for NSCLC, its long-term oncological efficacies have been reported by a mere few studies. Yang et al. and Shagabayeva et al. found that RATL achieved a similar 5-year survival to VATL and OL for stage I and II-IIIA NSCLC, respectively [41,42]. Meanwhile, Herb et al. reported that RATL offered a long-term advantage over OL for clinical-N2 NSCLC patients, while a prospective study reported by Huang et al. did not show such superiority [14,15]. In the present study, we found that RATL achieved comparable oncological outcomes to VATL and OL for the treatment of N1 NSCLC. Meanwhile, the increased number of metastatic N1 LNs was the independent risk factor of poor DFS and OS for N1 NSCLC patients, while receiving adjuvant therapy could prolonged survival. However, the tumor size and visceral pleural invasion were not independently associated with the oncological outcomes. These results suggested that N1 lymph nodal metastasis was a more important factor than the primary tumor that impacts the oncological outcomes of N1 NSCLC patients. However, our recruitment ended in June 2020 and many cases have not reached a 5-year follow-up. The first RATL in the mainland of China was performed by our surgical team in 2009; further, our surgeons were unskilled when we first performed this innovative surgical technique. Therefore, many of the included patients in this study received RATL in recent years. In addition, NSCLC cases that underwent RATL, VATL, or OL in the same period were identified to minimize the potential bias attributed to the operation date. As a consequence, the 5-year follow-up profiles were available for a few cases. Nevertheless, most patients have finished the 3-year follow-up, and the results indicated that the three groups had close 1-year and 3-year survival data. Currently, we are proceeding with follow-up and are including more eligible patients, intending to further investigate the long-term outcomes of RATL, VATL, and OL. Moreover, the timing and site-specific recurrence patterns, which might provide better postoperative surveillance strategies for N1 NSCLC patients among the three surgical methods, may also require further detection.

To the best of our knowledge, this study is the largest retrospective study assessing the superiority of RATL over traditional surgical approaches regarding perioperative and oncological outcomes specified for NSCLC patients with N1 lymph nodal metastasis. Moreover, we also notice there are still some limitations of our study. Firstly, the present study is a retrospective study, which was, therefore, likely to result in undiscovered confounding and selection bias among the three groups. Although consecutive cases were included and PSM was applied to minimize the bias, participants were not randomized before operations, and a large percentage of patients who received VATL or OL were excluded after PSM. Therefore, randomized, controlled trials might be essential to further validate the conclusions of this study. Secondly, our research was conducted in one high-patient-volume medical center in China, which restricted the representation of the participants. Therefore, further national multi-center studies are needed to ensure the representation of this study. Finally, our study merely enrolled N1 NSCLC patients who received lobectomy, and further studies evaluating the feasibility and efficacy of RATL for other resectable advanced-stage NSCLC (e.g., T3-4 or N2 NSCLC) are necessary to expand the application of RATL in treating NSCLC.

## 5. Conclusions

Our study demonstrated that RATL resulted in the optimal surgical outcomes, the fastest recovery, and the lowest morbidities of postsurgical complications among the three surgical methods that were analyzed. Further, it also showed superiority in the N1 LN assessment over VATL, though it achieved comparable oncological outcomes in regard to VATL and OL for NSCLC patients with N1 lymph nodal metastasis. Further follow-up and prospective studies are necessary in order to verify our findings. 

## Figures and Tables

**Figure 1 cancers-14-05249-f001:**
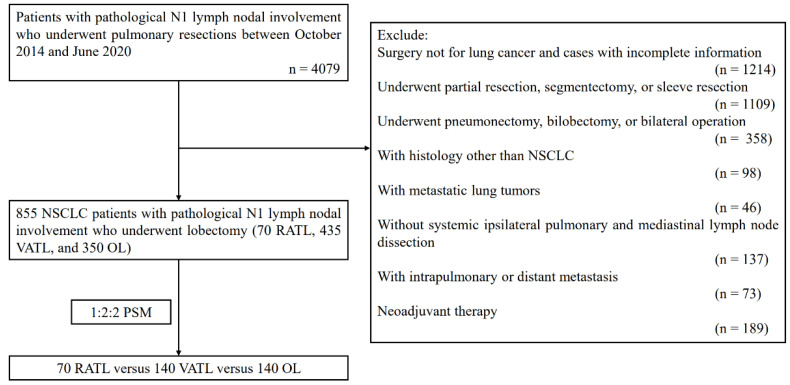
Flow chart of the study population. NSCLC: non-small cell lung cancer; RATL: robotic-assisted thoracoscopic lobectomy; VATL: video-assisted thoracoscopic lobectomy; OL: open lobectomy; and PSM: propensity score matching.

**Figure 2 cancers-14-05249-f002:**
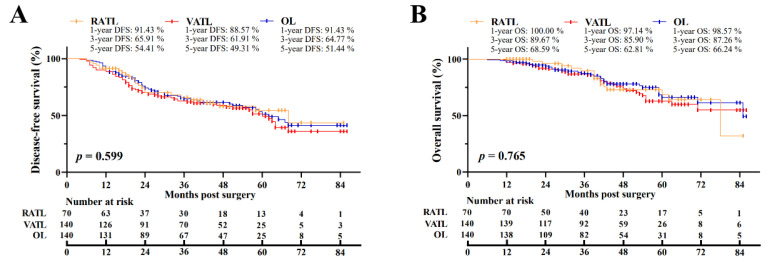
Oncological outcomes of N1 NSCLC patients. The comparison of disease-free (**A**) and overall (**B**) survival among the RATL, VATL, and OL groups. RATL: robotic-assisted thoracoscopic lobectomy; VATL: video-assisted thoracoscopic lobectomy; and OL: open lobectomy.

**Figure 3 cancers-14-05249-f003:**
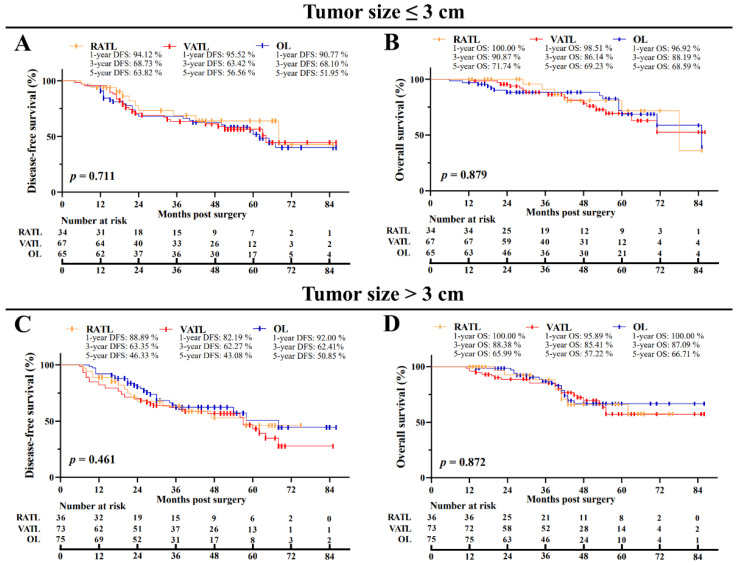
Subgroup analysis of oncological outcomes of N1 NSCLC patients in terms of the tumor size. The comparison of disease-free (**A**) and overall (**B**) survival for cases with a tumor size ≤ 3.0 cm. The comparison of disease-free (**C**) and overall (**D**) survival for cases with a tumor size > 3.0 cm. RATL: robot-assisted thoracoscopic lobectomy; VATL: video-assisted thoracoscopic lobectomy; and OL: open lobectomy.

**Table 1 cancers-14-05249-t001:** Baseline characteristics and pathologic details of unmatched patients.

Variables	RATL (n = 70)	VATL (n = 435)	OL (n = 350)	*p* Value
Age (years), mean ± SD	59.23 ± 10.48	60.63 ± 9.27	60.70 ± 8.18	0.621
Sex, n (%) Male Female	45 (64.29) 25 (35.71)	290 (66.67) 145 (33.33)	285 (81.43) 65 (18.57)	<0.001
Smoking status, n (%) Never Former Active	32 (45.71) 13 (18.57) 25 (35.71)	201 (46.21) 84 (19.31) 150 (34.48)	131 (37.43) 46 (13.14) 173 (49.43)	<0.001
BMI (kg/m^2^), mean ± SD	24.07 ± 3.29	23.88 ± 3.22	23.54 ± 2.96	0.344
Diabetes mellitus, n (%)	6 (8.57)	35 (8.05)	29 (8.29)	0.985
Coronary artery disease, n (%)	3 (4.29)	15 (3.45)	15 (4.29)	0.817
Hypertension, n (%)	17 (24.29)	118 (27.13)	95 (27.14)	0.876
Chronic obstructive pulmonary disease, n (%)	1 (1.43)	14 (3.22)	19 (5.43)	0.142
FEV1 (% of predicted), mean ± SD	92.79 ± 12.83	90.16 ± 16.25	90.98 ± 16.72	0.464
DLCO (% of predicted), mean ± SD	97.10 ± 17.40	97.11 ± 20.86	97.72 ± 18.76	0.752
History of malignancy, n (%)	1 (1.43)	5 (1.15)	4 (1.14)	0.978
Tumor location, n (%) Right upper lobe Right middle lobe Right lower lobe Left upper lobe Left lower lobe	16 (22.86) 11 (15.71) 20 (28.57) 7 (10.00) 16 (22.86)	111 (25.52) 40 (9.20) 92 (21.15) 110 (25.29) 82 (18.85)	96 (27.43) 39 (11.14) 65 (18.57) 85 (24.29) 65 (18.57)	0.118
Histology, n (%) Adenocarcinoma Squamous cell Mixed/large cell/others	57 (81.43) 9 (12.86) 4 (5.71)	335 (77.01) 74 (17.01) 26 (5.98)	123 (35.14) 207 (59.14) 20 (5.71)	<0.001
Tumor size (cm), mean ± SD	3.07 ± 1.03	3.07 ± 1.42	4.65 ± 1.91	<0.001
Visceral pleural invasion, n (%)	22 (31.43)	153 (35.17)	106 (30.29)	0.334
Pathological T stage, n (%) T1 T2 T3 T4	18 (25.71) 40 (57.14) 10 (14.29) 2 (2.86)	127 (29.20) 228 (52.41) 70 (16.09) 10 (2.30)	64 (18.29) 167 (47.71) 79 (22.57) 40 (11.43)	<0.001
Pathological TNM stage, n (%) Stage IIB Stage IIIA	58 (82.86) 12 (17.14)	357 (82.07) 78 (17.93)	231 (66.00) 119 (34.00)	<0.001
Clinical N stage, n (%) N0 N1 N2	48 (68.57) 16 (22.86) 6 (8.57)	259 (59.54) 117 (26.90) 59 (13.56)	198 (56.57) 88 (25.14) 64 (18.29)	0.138
Adjuvant therapy, n (%) Chemotherapy Chemoradiotherapy Others ^a^ None	58 (82.86) 4 (5.71) 3 (4.29) 5 (7.14)	340 (78.16) 29 (6.67) 31 (7.13) 35 (8.05)	290 (82.86) 23 (6.57) 11 (3.14) 26 (7.43)	0.350

Categorical variables are expressed in a number (percentage), and continuous variables are shown as mean ± SD. ^a^: immunotherapy or target therapy; RATL: robotic-assisted thoracoscopic lobectomy; VATL: video-assisted thoracoscopic lobectomy; OL: open lobectomy; BMI: body mass index; FEV1: forced expiratory volume in 1 s; and DLCO: diffusing capacity for carbon monoxide.

**Table 2 cancers-14-05249-t002:** Baseline characteristics and pathologic details of matched patients.

Variables	RATL (n = 70)	VATL (n = 140)	OL (n = 140)	*p* Value
Age (years), mean ± SD	59.23 ± 10.48	59.67 ± 9.49	59.30 ± 8.66	0.986
Sex, n (%) Male Female	45 (64.29) 25 (35.71)	85 (60.71) 55 (39.29)	91 (65.00) 49 (35.00)	0.740
Smoking status, n (%) Never Former Active	32 (45.71) 13 (18.57) 25 (35.71)	68 (48.57) 21 (15.00) 51 (36.43)	63 (45.00) 22 (15.71) 55 (39.29)	0.937
BMI (kg/m^2^), mean ± SD	24.07 ± 3.29	24.31 ± 3.08	23.68 ± 2.89	0.426
Diabetes mellitus, n (%)	6 (8.57)	14 (10.00)	11 (7.86)	0.816
Coronary artery disease, n (%)	3 (4.29)	4 (2.86)	7 (5.00)	0.682
Hypertension, n (%)	17 (24.29)	38 (27.14)	39 (27.86)	0.855
Chronic obstructive pulmonary disease, n (%)	1 (1.43)	4 (2.86)	5 (3.57)	0.780
FEV1 (% of predicted), mean ± SD	92.79 ± 12.83	90.96 ± 15.16	94.99 ± 15.50	0.396
DLCO (% of predicted), mean ± SD	97.10 ± 17.40	97.26 ± 19.87	97.68 ± 16.99	0.637
History of malignancy, n (%)	1 (1.43)	2 (1.43)	1 (0.71)	1.000
Tumor location, n (%) Right upper lobe Right middle lobe Right lower lobe Left upper lobe Left lower lobe	16 (22.86) 11 (15.71) 20 (28.57) 7 (10.00) 16 (22.86)	29 (20.71) 18 (12.86) 28 (20.00) 29 (20.71) 36 (25.71)	35 (25.00) 27 (19.29) 28 (20.00) 19 (13.57) 31 (22.14)	0.363
Histology, n (%) Adenocarcinoma Squamous cell Mixed/large cell/others	57 (81.43) 9 (12.86) 4 (5.71)	118 (84.29) 16 (11.43) 6 (4.29)	103 (73.57) 29 (20.71) 8 (5.71)	0.228
Tumor size (cm), mean ± SD	3.07 ± 1.03	3.02 ± 1.01	3.16 ± 1.09	0.407
Visceral pleural invasion, n (%)	22 (31.43)	39 (27.86)	34 (24.29)	0.531
Pathological T stage, n (%) T1 T2 T3 T4	18 (25.71) 40 (57.14) 10 (14.29) 2 (2.86)	40 (28.57) 75 (53.57) 22 (15.71) 3 (2.14)	36 (25.71) 73 (52.14) 26 (18.57) 5 (3.57)	0.957
Pathological TNM stage, n (%) Stage IIB Stage IIIA	58 (82.86) 12 (17.14)	115 (82.14) 25 (17.86)	113 (80.71) 27 (19.29)	0.918
Clinical N stage, n (%) N0 N1 N2	48 (68.57) 16 (22.86) 6 (8.57)	76 (54.29) 47 (33.57) 17 (12.14)	94 (67.14) 31 (22.14) 15 (10.71)	0.142
Adjuvant therapy, n (%) Chemotherapy Chemoradiotherapy Others ^a^ None	58 (82.86) 4 (5.71) 3 (4.29) 5 (7.14)	110 (78.57) 12 (8.57) 6 (4.29) 12 (8.57)	113 (80.71) 9 (6.43) 5 (3.57) 13 (9.29)	0.985

Categorical variables are expressed as number (percentage), and continuous variables are shown as mean ± SD. ^a^: immunotherapy or target therapy; RATL: robotic-assisted thoracoscopic lobectomy; VATL: video-assisted thoracoscopic lobectomy; OL: open lobectomy; BMI: body mass index; FEV1: forced expiratory volume in 1 s; and DLCO: diffusing capacity for carbon monoxide.

**Table 3 cancers-14-05249-t003:** Perioperative outcomes.

Characteristics	RATL (n = 70)	VATL (n = 140)	OL (n = 140)	*p* Value	RATL vs. VATL ^a^	RATL vs. OL ^b^
Resection margin, n (%) R0 ^c^ R1 ^d^ R2 ^e^	68 (97.14) 2 (2.86) 0 (0.00)	132 (94.29) 8 (5.71) 0 (0.00)	128 (91.43) 9 (6.43) 3 (2.14)	0.330	>0.050	>0.050
Reoperation, n (%)	1 (1.43)	4 (2.86)	5 (3.57)	0.780	>0.050	>0.050
Surgical time (mins), mean ± SD	93.37 ± 35.85	105.53 ± 32.33	102.84 ± 35.11	0.005	0.003	0.085
Intraoperative Blood loss, n (%) ≤100 mL >100 mL	60 (85.71) 10 (14.29)	117 (83.57) 23 (16.43)	79 (56.43) 61 (43.57)	<0.001	>0.050	<0.050
Intraoperative blood transfusion, n (%)	1 (1.43)	1 (0.71)	3 (2.14)	0.844	>0.050	>0.050
ICU stay (days), median(range)	0(0–2)	0(0–5)	1(0–11)	<0.001	0.394	<0.001
Chest tube drainage, median(range) Duration (days) Volume (mL)	4(1–17) 800(220–1590)	4(2–25) 775(200–2800)	5(2–23) 800(220–2500)	<0.001 0.967	1.000 1.000	<0.001 1.000
Postsurgical stay (days), median(range)	5(2–18)	5(2–27)	6(2–28)	<0.001	0.139	<0.001
Postsurgical complications, n (%)	7 (10.00)	20 (14.29)	34 (24.29)	0.016	>0.050	<0.050
Pneumonia requiring antibiotics	3 (4.29)	4 (2.86)	10 (7.14)	0.230	>0.050	>0.050
Bronchopleural fistula	0 (0.00)	1 (0.71)	3 (2.14)	0.536	-	-
Atrial fibrillation	0 (0.00)	0 (0.00)	2 (1.43)	0.358	-	-
Recurrent laryngeal nerve injury	0 (0.00)	1 (0.71)	1 (0.71)	1.000	-	-
Hemorrhage requiring intervention	1 (1.43)	2 (1.43)	3 (2.14)	1.000	>0.050	>0.050
Prolonged air leak > 5 days	5 (7.14)	12 (8.57)	18 (12.86)	0.329	>0.050	>0.050
Chylothorax	1 (1.43)	2 (1.43)	3 (2.14)	1.000	>0.050	>0.050
Pyothorax	0 (0.00)	0 (0.00)	1 (0.71)	1.000	-	-
Subcutaneous emphysema	4 (5.71)	10 (7.14)	8 (5.71)	0.884	>0.050	>0.050
Wound infection	1 (1.43)	1 (0.71)	2 (1.43)	1.000	>0.050	>0.050
ARDS	0 (0.00)	1 (0.71)	1 (0.71)	1.000	-	-
Chest tube reinsertion	1 (1.43)	1 (0.71)	2 (1.43)	1.000	>0.050	>0.050
In-hospital mortality	0 (0.00)	0 (0.00)	0 (0.00)	-	-	-
Readmission	0 (0.00)	1 (0.71)	2 (1.43)	0.806	-	-
30 d mortality	0 (0.00)	0 (0.00)	0 (0.00)	-	-	-

Categorical variables are expressed as number (percentage), and continuous variables are shown as mean ± SD or median(range). ^a^: Adjusted *p* value of multiple comparisons between the RATL and VATL group; ^b^: adjusted *p* value of multiple comparisons between the RATL and OL group; ^c^: no residual tumor; ^d^: residual microscopic tumor; and ^e^: residual macroscopic tumor. RATL: robotic-assisted thoracoscopic lobectomy; VATL: video-assisted thoracoscopic lobectomy; OL: open lobectomy; and ARDS: acute respiratory distress syndrome.

**Table 4 cancers-14-05249-t004:** LNs assessment.

Variables	RATL (n = 70)	VATL (n = 140)	OL (n = 140)	*p* Value	RATL vs. VATL ^a^	RATL vs. OL ^b^
N1 LNs dissected Overall count, mean ± SD Positive count, mean ± SD	7.01 ± 1.77 1.84 ± 1.36	6.21 ± 2.30 1.76 ± 1.03	7.19 ± 3.41 1.89 ± 1.02	0.013 0.363	0.027 1.000	1.000 0.473
Number of #10 LNs dissected Frequency assessed, n (%) Overall count, mean ± SD Positive count, mean ± SD	62 (88.57) 1.97 ± 1.22 0.37 ± 0.81	116 (82.86) 1.68 ± 1.18 0.34 ± 0.52	124 (88.57) 2.04 ± 1.46 0.39 ± 0.61	0.314 0.078 0.369	>0.050 0.220 0.804	>0.050 1.000 0.493
Number of #11 LNs dissected Frequency assessed, n (%) Overall count, mean ± SD Positive count, mean ± SD	67 (95.71) 2.53 ± 1.24 0.46 ± 0.76	128 (91.43) 2.35 ± 1.31 0.43 ± 0.62	134 (95.71) 2.61 ± 1.70 0.54 ± 0.53	0.326 0.553 0.286	>0.050 0.916 0.947	>0.050 1.000 0.610
Number of #12 LNs dissected Frequency assessed, n (%) Overall count, mean ± SD Positive count, mean ± SD	60 (85.71) 2.51 ± 1.67 1.01 ± 1.35	107 (76.43) 2.19 ± 1.81 0.99 ± 0.90	119 (85.00) 2.54 ± 2.17 0.96 ± 1.00	0.130 0.248 0.567	>0.050 0.318 0.917	>0.050 1.000 1.000
N1 stations dissected Overall count, mean ± SD	2.70 ± 0.49	2.62 ± 0.56	2.69 ± 0.49	0.521	1.000	1.000
Positive count, mean ± SD	1.26 ± 0.44	1.29 ± 0.48	1.31 ± 0.58	0.962	1.000	1.000
N2 LNs count	8.83 ± 4.09	7.88 ± 3.30	8.86 ± 4.18	0.225	0.645	1.000
N2 stations count, mean ± SD	3.71 ± 1.02	3.61 ± 1.04	3.72 ± 1.15	0.556	1.000	1.000
N1 + N2 LNs count, mean ± SD	15.84 ± 4.49	14.09 ± 4.09	16.05 ± 5.17	0.001	0.027	0.950
N1 + N2 stations count, mean ± SD	6.41 ± 1.10	6.23 ± 1.19	6.41 ± 1.19	0.367	0.953	1.000
Postoperative upstaging (cN0-pN1), n (%)	48 (68.57)	76 (54.29)	94 (67.14)	0.040	<0.050	>0.050

Categorical variables are expressed as number (percentage), and continuous variables are shown as mean ± SD. ^a^: Adjusted *p* value of multiple comparisons between the RATL and VATL groups and ^b^: adjusted *p* value of multiple comparisons between the RATL and OL group. RATL: robotic-assisted thoracoscopic lobectomy; VATL: video-assisted thoracoscopic lobectomy; OL: open lobectomy; and LNs: lymph nodes.

**Table 5 cancers-14-05249-t005:** Multivariate Cox regression model analysis for oncological outcomes.

Predictors of Survival	Disease-Free Survival			Overall Survival		
	*p* Value	Hazard Ratio	95% CI	*p* Value	Hazard Ratio	95% CI
Surgical type (RATL vs. others)	0.970	1.008	0.645–1.576	0.577	0.855	0.492–1.485
Age (<60 vs. ≥60 years)	0.484	0.888	0.636–1.239	0.502	0.862	0.559–1.329
Smoking history (yes vs. no)	0.782	0.954	0.685–1.330	0.507	1.155	0.755–1.766
Sex (male vs. female)	0.434	0.870	0.613–1.234	0.683	0.911	0.582–1.426
Histologic subtype (ADC vs. SCC)	0.386	1.499	0.600–3.743	0.516	1.480	0.453–4.836
Tumor size (≤3 vs. >3 cm)	0.638	0.923	0.661–1.289	0.879	0.967	0.628–1.488
Visceral pleural invasion (yes vs. no)	0.058	0.714	0.504–1.012	0.325	0.796	0.506–1.253
Number of positive LNs (1 vs. >1)	<0.001	0.268	0.184–0.390	<0.001	0.368	0.232–0.582
Adjuvant therapy (yes vs. no)	0.012	0.520	0.312–0.867	0.003	0.405	0.225–0.731

CI: confidence interval; RATL: robot-assisted thoracoscopic surgery; SCC: squamous cell carcinoma; and ADC: adenocarcinoma.

## Data Availability

All data presented in this study are available upon reasonable request from the corresponding author Qingquan Luo.

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
