# Peer review of "Perioperative and Oncological Outcomes of Robotic-Assisted, Video-Assisted Thoracoscopic and Open Lobectomy for Patients with N1-Metastatic Non-Small Cell Lung Cancer: A Propensity Score-Matched Study"

_cancers, 2022, doi:10.3390/cancers14215249_

Round 1
Reviewer 1 Report
This is an interesting paper on surgical and oncological outcomes of robotic lobectomy in pN1 NSCLC, compared to VATS and open surgery. In general terms, it is well-designed, illustrated and referenced.
The authors reported as inclusion criteria pathologic N1. Can the clinical lymph nodal staging specify in the results? The rate of cN0 in the 3 groups could influence the different N1 nodal upstaging
The section “results” appears too detailed, with a series of data existing in the tables. Please, resume the results, to make the reading more fluent. Generally, it is not needed to repeat the information in the table also in the text
Author Response
Dear Reviewer,
Re: Manuscript ID: cancers-1970220
On behalf of my co-authors, we sincerely appreciate you for the constructive comments and suggestions on our manuscript entitled “Perioperative and Oncological Outcomes of Robotic-assisted, Video-assisted Thoracoscopic and Open Lobectomy for Patients with N1-metastatic Non-small Cell Lung Cancer: A Propensity Score-Matched Study (Manuscript ID: cancers-1970220)”. In our revised manuscript, we have dealt with all the comments in a point-by-point fashion per your instructions and all the revisions were highlighted in red. The revised manuscript is hereby attached along with the responses to the comments attached below for your review. We sincerely hope that the revision meets your satisfaction. Please see the attachment.
Yours truly,
Qingquan Luo, MD
Professor, Corresponding author
Department of Thoracic Surgical Oncology, Shanghai Lung Cancer Center, Shanghai Jiao Tong University School of Medicine
Shanghai, China
Email: luoqingquan@hotmail.com

Reviewer 2 Report
Dear Authors,
Thank you for your work on investigating the utility of robotic lung surgery as it pertains to patients with NSCLC with N1 disease.
1) For Table 2 which represents the matched patients, the N numbers in the table need to be corrected.
2) The study identified patients based on having pathologic N1 disease as a criteria for enrollment. Is there a reason that clinical stage was not used? Based on the methods, the patients all had standard staging (CT/PET/MRI) imaging and procedures (EBUS/mediastinoscopy). By using pathologic N1 patients, you introduce significant selection bias. For example, patients in the robotic cohort could have clinical 1A disease and were upstaged to stage 2 whereas patients in the open group may have known N1 disease and were selected to undergo an open operation due to anatomic concerns of advanced disease. This is evident by the fact that in both the unmatched and matched groups, there is a significant difference between the percentage of stage 3A and 2B patients between the RATS and OL groups (17% vs 34%-unmatched and 17% vs. 24%-matched). This also leads to the somewhat surprising finding that DFS and OS are equal between the groups. With a higher proportion of stage 3A patients in the OL group, one would have predicted a worse 3yr and 5yr DFS and OS. However, would the fact that they are equal raise the question of whether RATS may be inferior to OL from an oncologic perspective? Perhaps, as a retrospective study, there would be less bias if the initial sample was based on clinical stage -- for example, patients with clinical stage 2A & 2B disease who underwent RATS, VATS, OL. You then can have a better estimate for upstaging to stage 3A and a better comparison on survival and DFS between the groups.
Author Response

(The authors gave the same response as above.)

Round 2
Reviewer 2 Report
Thank you for your comments and revisions. Appreciate the additional information on clinical stage and corrections to the data.